# DATNet: Dual Adversarial Transfer for Low-resource Named Entity Recognition

## Abstract

We propose a new architecture termed Dual Adversarial Transfer Network (DATNet) for addressing low-resource Named Entity Recognition (NER). Specifically, two variants of DATNet, i.e., DATNet-F and DATNet-P, are proposed to explore effective feature fusion between high and low resource. To address the noisy and imbalanced training data, we propose a novel Generalized Resource-Adversarial Discriminator (GRAD). Additionally, adversarial training is adopted to boost model generalization. We examine the effects of different components in DATNet across domains and languages, and show that significant improvement can be obtained especially for low-resource data. Without augmenting any additional hand-crafted features, we achieve new state-of-the-art performances on CoNLL and Twitter NER—88.16% F1 for Spanish, 53.43% F1 for WNUT-2016, and 42.83% F1 for WNUT-2017[1].

## 1 Introduction

Named entity recognition (NER) is an important step in most natural language processing (NLP) applications. It detects not only the type of named entity, but also the entity boundaries, which requires deep understanding of the contextual semantics to disambiguate the different entity types of same tokens. To tackle this challenging problem, most early studies were based on hand-crafted rules, which suffered from limited performance in practice. Current methods are devoted to developing learning based algorithms, especially neural network based methods, and have been advancing the state-of-the-art consecutively (Collobert et al., 2011; Huang et al., 2015; Lample et al., 2016; Chiu & Nichols, 2016; Ma & Hovy, 2016). These end-to-end models generalize well on new entities based on features automatically learned from the data. However, when the annotated corpora is small, especially in the low resource scenario (Zhang et al., 2016), the performance of these methods degrades significantly since the hidden feature representations cannot be learned adequately.

Recently, more and more approaches have been proposed to address low-resource NER. Early works (Chen et al., 2010; Li et al., 2012) primarily assumed a large parallel corpus and focused on exploiting them to project information from high- to low-resource. Unfortunately, such a large parallel corpus may not be available for many low-resource languages. More recently, cross-resource word embedding (Fang & Cohn, 2017; Adams et al., 2017; Yang et al., 2017) was proposed to bridge the low and high resources and enable knowledge transfer. Although the aforementioned transfer-based methods show promising performance in low-resource NER, there are two issues deserved to be further investigated on: 1) Representation Difference - they did not consider the representation difference across resources and enforced the feature representation to be shared across languages/domains; 2) Resource Data Imbalance - the training size of high-resource is usually much larger than that of low-resource. The existing methods neglect such difference in their models, resulting in poor generalization.

In this work, we present an approach termed **Dual Adversarial Transfer Network (DATNet)** to address the above issues in a unified framework for low-resource NER. Specifically, to handle the representation difference, we first investigate on two architectures of hidden layers (we use bi-directional long-short term memory (BiLSTM) model as hidden layer) for transfer. The first one is that all the units in hidden layers are common units shared across languages/domains. The second

---

[1]The implementation details will be available at `https://github.com/` after acceptance.

one is composed of both private and common units, where the private part preserves the independent language/domain information. Extensive experiments are conducted to show their advantages over each other in different situations. On top of common units, the adversarial discriminator (AD) loss is introduced to encourage the resource-agnostic representation so that the knowledge from high resource can be more compatible with low resource. To handle the resource data imbalance issue, we further propose a variant of the AD loss, termed *Generalized Resource-Adversarial Discriminator (GRAD)*, to impose the resource weight during training so that low-resource and hard samples can be paid more attention to. In addition, we create adversarial samples to conduct the *Adversarial Training (AT)*, further improving the generalization and alleviating over-fitting problem. We unify two kinds of adversarial learning, i.e., GRAD and AT, into one transfer learning model, termed Dual Adversarial Transfer Network (DATNet), to achieve end-to-end training and obtain the state-of-the-art performance on a series of NER tasks–88.16% F1 for CoNLL-2002 Spanish, 53.43% and 42.83% F1 for WNUT-2016 and 2017. Different from prior works, we do *not* use additional hand-crafted features and do *not* use cross-lingual word embeddings while addressing the cross-language tasks.

## 2 RELATED WORK

**Named Entity Recognition**    NER is typically framed as a sequence labeling task which aims at automatic detection of named entities (e.g., person, organization, location and etc.) from free text (Marrero et al., 2013). The early works applied CRF, SVM, and perception models with hand-crafted features (Ratinov & Roth, 2009; Passos et al., 2014; Luo et al., 2015). With the advent of deep learning, research focus has been shifting towards deep neural networks (DNN), which requires little feature engineering and domain knowledge (Lample et al., 2016; Zukov Gregoric et al., 2018). Collobert et al. (2011) proposed a feed-forward neural network with a fixed sized window for each word, which failed in considering useful relations between long-distance words. To overcome this limitation, Chiu & Nichols (2016) presented a bidirectional LSTM-CNNs architecture that automatically detects word- and character-level features. Ma & Hovy (2016) further extended it into bidirectional LSTM-CNNs-CRF architecture, where the CRF module was added to optimize the output label sequence. Liu et al. (2018) proposed task-aware neural language model termed LM-LSTM-CRF, where character-aware neural language models were incorporated to extract character-level embedding under a multi-task framework.

**Transfer Learning for NER**    Transfer learning can be a powerful tool to low resource NER tasks. To bridge high and low resource, transfer learning methods for NER can be divided into two types: the parallel corpora based transfer and the shared representation based transfer. Early works mainly focused on exploiting parallel corpora to project information between the high- and low-resource language (Yarowsky et al., 2001; Chen et al., 2010; Li et al., 2012; Feng et al., 2018). For example, Chen et al. (2010) and Feng et al. (2018) proposed to jointly identify and align bilingual named entities. On the other hand, the shared representation methods do not require the parallel correspondence (Rei & Søgaard, 2018). For instance, Fang & Cohn (2017) proposed cross-lingual word embeddings to transfer knowledge across resources. Yang et al. (2017) presented a transfer learning approach based on a deep hierarchical recurrent neural network (RNN), where full/partial hidden features between source and target tasks are shared. Ni *et al.* (Ni & Florian, 2016; Ni et al., 2017) utilized the Wikipedia entity type mappings to improve low-resource NER. Al-Rfou' et al. (2015) built massive multilingual annotators with minimal human expertise by using language agnostic techniques. Mayhew et al. (2017) created a cross-language NER system, which works well for very minimal resources by translate annotated data of high-resource into low-resource. Cotterell & Duh (2017) proposed character-level neural CRFs to jointly train and predict low- and high-resource languages. Pan et al. (2017) proposes a large-scale cross-lingual named entity dataset which contains 282 languages for evaluation. In addition, multi-task learning (Yang et al., 2016; Luong et al., 2016; Rei, 2017; Aguilar et al., 2017; Hashimoto et al., 2017; Lin et al., 2018) shows that jointly training on multiple tasks/languages helps improve performance. Different from transfer learning methods, multi-task learning aims at improving the performance of all the resources instead of low resource only.

**Adversarial Learning**    Adversarial learning originates from Generative Adversarial Nets (GAN) (Goodfellow et al., 2014), which shows impressing results in computer vision. Recently, many papers have tried to apply adversarial learning to NLP tasks. Liu et al. (2017) presented an adversarial multi-task learning framework for text classification. Gui et al. (2017) applied the adversarial

discriminator to POS tagging for Twitter. Kim et al. (2017) proposed a language discriminator to enable language-adversarial training for cross-language POS tagging. Apart from adversarial discriminator, adversarial training is another concept originally introduced by (Szegedy et al., 2014; Goodfellow et al., 2015) to improve the robustness of image classification model by injecting malicious perturbations into input images. Recently, Miyato et al. (2017) proposed a semi-supervised text classification method by applying adversarial training, where for the first time adversarial perturbations were added onto word embeddings. Yasunaga et al. (2018) applied adversarial training to POS tagging. Different from all these adversarial learning methods, our method integrates both the adversarial discriminator and adversarial training in an unified framework to enable end-to-end training.

# 3 DUAL ADVERSARIAL TRANSFER NETWORK (DATNET)

In this section, we introduce DATNet in more details. We first describe a base model for NER, and then discuss two proposed transfer architectures for DATNet.

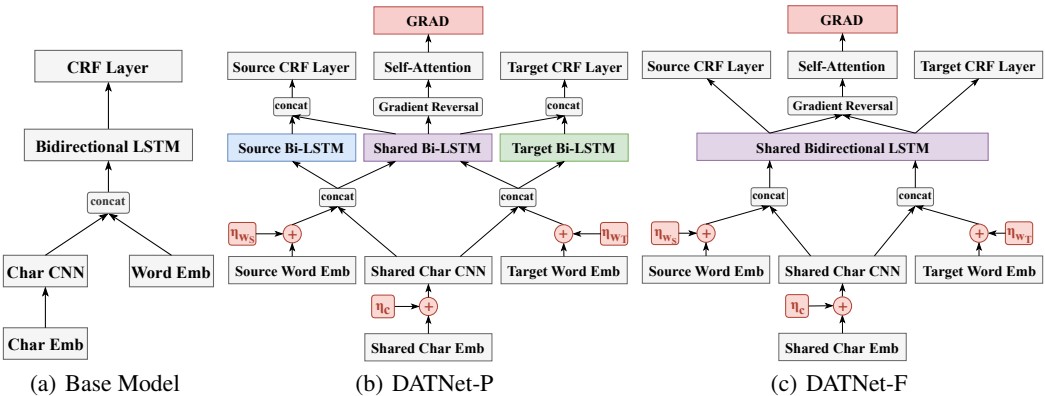

Figure 1: The general architecture of proposed models.

## 3.1 BASIC ARCHITECTURE

We follow state-of-the-art models for NER task (Huang et al., 2015; Lample et al., 2016; Chiu & Nichols, 2016; Ma & Hovy, 2016), i.e., LSTM-CNNs-CRF based structure, to build the base model. It consists of the following pieces: character-level embedding, word-level embedding, BiLSTM for feature representation, and CRF as the decoder. The character-level embedding takes a sequence of characters in the word as atomic units input to derive the word representation that encodes the morphological information, such as root, prefix, and suffix. These character features are usually encoded by character-level CNN or BiLSTM, then concatenated with word-level embedding to form the final word vectors. On top of them, the network further incorporates the contextual information using BiLSTM to output new feature representations, which is subsequently fed into CRF layer to predict label sequence. Although both of the word-level layer and the character-level layer can be implemented using CNNs or RNNs, we use CNNs for extracting character-level and RNNs for extracting word-level representation. Fig. 1(a) shows the the architecture of the base model.

## 3.2 DUAL ADVERSARIAL TRANSFER ARCHITECTURE

### 3.2.1 CHARACTER-LEVEL ENCODER

Previous works have shown that character features can boost sequence labeling performance by capturing morphological and semantic information (Lin et al., 2018). For low-resource dataset to obtain high-quality word features, character features learned from other language/domain may provide crucial information for labeling, especially for rare and out-of-vocabulary words. Character-level encoder usually contains BiLSTM (Lample et al., 2016) and CNN (Chiu & Nichols, 2016; Ma & Hovy,

2016) approaches. In practice, Reimers & Gurevych (2017) observed that the difference between the two approaches is statistically insignificant in sequence labeling tasks, but character-level CNN is more efficient and has less parameters. Thus, we use character-level CNN and share character features between high- and low-resource tasks to enhance the representations of low-resource.

### 3.2.2 WORD-LEVEL ENCODER

To learn a better word-level representation, we concatenate character-level features of each word with a latent word embedding as $\mathbf{w}_i = [\mathbf{w}_i^{char}, \mathbf{w}_i^{emb}]$, where the latent word embedding $\mathbf{w}_i^{emb}$ is initialized with pre-trained embeddings and fixed during training. One unique characteristic of NER is that the historical and future input for a given time step could be useful for label inference. To exploit such a characteristic, we use a bidirectional LSTM architecture (Hochreiter & Schmidhuber, 1997) to extract contextualized word-level features. In this way, we can gather the information from the past and future for a particular time frame $t$ as follows, $\overrightarrow{\mathbf{h}}_t = \texttt{lstm}(\overrightarrow{\mathbf{h}}_{t-1}, \mathbf{w}_t)$, $\overleftarrow{\mathbf{h}}_t = \texttt{lstm}(\overleftarrow{\mathbf{h}}_{t+1}, \mathbf{w}_t)$. After the LSTM layer, the representation of a word is obtained by concatenating its left and right context representation as follows, $\mathbf{h}_t = [\overrightarrow{\mathbf{h}}_t, \overleftarrow{\mathbf{h}}_t]$.

To consider the resource representation difference on word-level features, we introduce two kinds of transferable word-level encoder in our model, namely DATNet-Full Share (DATNet-F) and DATNet-Part Share (DATNet-P). In DATNet-F, all the BiLSTM units are shared by both resources while word embeddings for different resources are disparate. The illustrative figure is depicted in the Fig. 1(c). Different from DATNet-F, the DATNet-P decomposes the BiLSTM units into the shared component and the resource-related one, which is shown in the Fig. 1(b).

### 3.2.3 GENERALIZED RESOURCE-ADVERSARIAL DISCRIMINATOR

In order to make the feature representation extracted from the source domain more compatible with those from the target domain, we encourage the outputs of the shared BiLSTM part to be resource-agnostic by constructing a resource-adversarial discriminator, which is inspired by the Language-Adversarial Discriminator proposed by Kim et al. (2017). Unfortunately, previous works did not consider the imbalance of training size for two resources. Specifically, the target domain consists of very limited labeled training data, e.g., 10 sentences. In contrast, labeled training data in the source domain are much richer, e.g., 10k sentences. If such imbalance was not considered during training, the stochastic gradient descent (SGD) optimization would make the model more biased to high resource (Lin et al., 2017b). To address this imbalance problem, we impose a weight $\alpha$ on two resources to balance their influences. However, in the experiment we also observe that the easily classified samples from high resource comprise the majority of the loss and dominate the gradient. To overcome this issue, we further propose Generalized Resource-Adversarial Discriminator (GRAD) to enable adaptive weights for each sample (note that the sample here means each sentence of resource), which focuses the model training on hard samples.

To compute the loss of GRAD, the output sequence of the shared BiLSTM is firstly encoded into a single vector via a self-attention module (Bahdanau et al., 2015), and then projected into a scalar $r$ via a linear transformation. The loss function of the resource classifier is formulated as:

$$\ell_{GRAD} = -\sum_i \{\mathbf{I}_{i \in \mathcal{D}_S} \alpha (1 - r_i)^\gamma \log r_i + \mathbf{I}_{i \in \mathcal{D}_T} (1 - \alpha) r_i^\gamma \log(1 - r_i)\} \tag{1}$$

where $\mathbf{I}_{i \in \mathcal{D}_S}, \mathbf{I}_{i \in \mathcal{D}_T}$ are the identity functions to denote whether a sentence is from high resource (source) and low resource (target), respectively; $\alpha$ is a weighting factor to balance the loss contribution from high and low resource; the parameter $(1 - r_i)^\gamma$ (or $r_i^\gamma$) controls the loss contribution from individual samples by measuring the discrepancy between prediction and true label (easy samples have smaller contribution); and $\gamma$ scales the contrast of loss contribution from hard and easy samples. In practice, the value of $\gamma$ does not need to be tuned much and usually set as 2 in our experiment. Intuitively, the weighting factors $\alpha$ and $(1 - r_i)^\gamma$ reduce the loss contribution from high resource and easy samples, respectively. Note that though the resource classifier is optimized to minimize the resource classification error, when the gradients originated from the resource classification loss are back-propagated to the other model parts than the resource classifier, they are negated for parameter updates so that these bottom layers are trained to be resource-agnostic.

### 3.2.4 Label Decoder

The label decoder induces a probability distribution over sequences of labels, conditioned on the word-level encoder features. In this paper, we use a linear chain model based on the first-order Markov chain structure, termed the chain conditional random field (CRF) Lafferty et al. (2001), as the decoder. In this decoder, there are two kinds of cliques: local cliques and transition cliques. Specifically, local cliques correspond to the individual elements in the sequence. And transition cliques, on the other hand, reflect the evolution of states between two neighboring elements at time $t-1$ and $t$ and we define the transition distribution as $\theta$. Formally, a linear-chain CRF can be written as $p(\mathbf{y}|\mathbf{h}_{1:T}) = \frac{1}{Z(\mathbf{h}_{1:T})} \exp\left\{\sum_{t=2}^{T} \theta_{y_{t-1},y_t} + \sum_{t=1}^{T} \mathbf{W}_{y_t} \mathbf{h}_t\right\}$, where $Z(\mathbf{h}_{1:T})$ is a normalization term and $\mathbf{y}$ is the sequence of predicted labels as follows: $\mathbf{y} = y_{1:T}$. Model parameters are optimized to maximize this conditional log likelihood, which acts as the objective function of the model. We define the loss function for source and target resources as follows, $\ell_S = -\sum_i \log p(\mathbf{y}|\mathbf{h}_{1:T})$, $\ell_T = -\sum_i \log p(\mathbf{y}|\mathbf{h}_{1:T})$.

### 3.2.5 Adversarial Training

So far our model can be trained end-to-end with standard back-propagation by minimizing the following loss:

$$\ell = \ell_{GRAD} + \ell_S + \ell_T \tag{2}$$

Recent works have demonstrated that deep learning models are fragile to *adversarial examples* Goodfellow et al. (2015). In computer vision, those adversarial examples can be constructed by changing a very small number of pixels, which are virtually indistinguishable to human perception (Pin-Yu et al., 2018). Recently, adversarial samples are widely incorporated into training to improve the generalization and robustness of the model, which is so-called adversarial training (AT) (Miyato et al., 2017). It emerges as a powerful regularization tool to stabilize training and prevent the model from being stuck in local minimum. In this paper, we explore AT in context of NER. To be specific, we prepare an adversarial sample by adding the original sample with a perturbation bounded by a small norm $\epsilon$ to maximize the loss function as follows:

$$\eta_{\mathbf{x}} = \arg \max_{\eta:\|\eta\|_2 \leq \epsilon} \ell(\Theta; \mathbf{x} + \eta) \tag{3}$$

where $\Theta$ is the current model parameters set. However, we cannot calculate the value of $\eta$ exactly in general, because the exact optimization with respect to $\eta$ is intractable in neural networks. Following the strategy in Goodfellow et al. (2015), this value can be approximated by linearizing it as follows,

$$\eta_{\mathbf{x}} = \epsilon \frac{\mathbf{g}}{\|\mathbf{g}\|_2}, \quad \text{where } \mathbf{g} = \nabla \ell(\Theta; \mathbf{x}) \tag{4}$$

where $\epsilon$ can be determined on the validation set. In this way, adversarial examples are generated by adding small perturbations to the inputs in the direction that most significantly increases the loss function of the model. We find such $\eta$ against the current model parameterized by $\Theta$, at each training step, and construct an adversarial example by $\mathbf{x}_{adv} = \mathbf{x} + \eta_{\mathbf{x}}$. Noted that we generate this adversarial example on the word and character embedding layer, respectively, as shown in the Fig. 1(b) and 1(c). Then, the classifier is trained on the mixture of original and adversarial examples to improve the generalization. To this end, we augment the loss in Eqn. 2 and define the loss function for adversarial training as:

$$\ell_{AT} = \ell(\Theta; \mathbf{x}) + \ell(\Theta; \mathbf{x}_{adv}) \tag{5}$$

where $\ell(\Theta; \mathbf{x}), \ell(\Theta; \mathbf{x}_{adv})$ represents the loss from an original example and its adversarial counterpart, respectively. Note that we present the AT in a general form for the convenience of presentation. For different samples, the loss and parameters should correspond to their counterparts. For example, for the source data with word embedding $\mathbf{w}_S$, the loss for AT can be defined as follows, $\ell_{AT} = \ell(\Theta; \mathbf{w}_S) + \ell(\Theta; \mathbf{w}_{S,adv})$ with $\mathbf{w}_{S,adv} = \mathbf{w}_S + \eta_{\mathbf{w}_S}$ and $\ell = \ell_{GRAD} + \ell_S$. Similarly, we can compute the perturbations $\eta_{\mathbf{c}}$ for char-embedding and $\eta_{\mathbf{w}_T}$ for target word embedding.

## 4 EXPERIMENTS

### 4.1 DATASETS

In order to evaluate the performance of DATNet, we conduct the experiments on following widely used NER datasets: CoNLL-2003 English NER (Kim & De, 2003), CoNLL-2002 Spanish & Dutch NER (Kim, 2002), WNUT-2016 & 2017 English Twitter NER (Zeman, 2017). The statistics of these datasets are described in Table 1. We use the official split of training/validation/test sets. Since our goal is to study the effects of transferring knowledge from high-resource dataset to low-resource dataset, unlike previous works (Collobert et al., 2011; Chiu & Nichols, 2016; Yang et al., 2017) to append one-hot gazetteer features to the input of the CRF layer, and the works (Partalas et al., 2016; Limsopatham & Collier, 2016; Aguilar et al., 2017) to introduce orthographic feature as additional input for learning social media NER in tweets, we do *not* experiment with hand-crafted features and only consider words and characters embeddings as the inputs of our model. To be noted, we used only train set for model training for all datasets except the WNUT-2016 NER dataset. Since in this dataset, all the previous studies merged the training and validation sets together for training, we followed the same way for fair comparison. Specifically, we use CoNLL-2003 English NER dataset as high-resource (i.e., source) for all the experiments on CoNLL and WNUT datasets, while CoNLL-2002 Spanish & Dutch NER datasets and WNUT-2016 & 2017 Twitter NER datasets as low-resource (i.e., target) in cross-language and cross-domain NER settings, respectively.

Table 1: Statistics of CoNLL and WNUT Named Entity Recognition Datasets.

| Benchmark | Resource | Language | # Training Tokens (# Entities) | # Dev Tokens (# Entities) | # Test Tokens (# Entities) |
|---|---|---|---|---|---|
| CoNLL-2003 | Source | English | 204,567 (23,499) | 51,578 (5,942) | 46,666 (5,648) |
| Cross-language NER | | | | | |
| CoNLL-2002 | Target | Spanish | 207,484 (18,797) | 51,645 (4,351) | 52,098 (3,558) |
| CoNLL-2002 | Target | Dutch | 202,931 (13,344) | 37,761 (2,616) | 68,994 (3,941) |
| Cross-domain NER | | | | | |
| WNUT-2016 | Target | English | 46,469 (2,462) | 16,261 (1,128) | 61,908 (5,955) |
| WNUT-2017 | Target | English | 62,730 (3,160) | 15,733 (1,250) | 23,394 (1,740) |

In addition to the CoNLL and WNUT datasets, we also experiment on the cross-language named entity dataset described in Pan et al. (2017), which contains datasets for 282 languages, to evaluate our methods and investigate the transferability of different linguistic families and branches in both low- and high-resource scenarios. We choose 9 languages in our experiment, where *Galician (gl)*, *West Frisian (fy)*, *Ukrainian (uk)* and *Marathi (mr)* are target languages, the corresponding source languages are *Spanish (es)*, *Dutch (nl)*, *Russian (ru)* and *Hindi (hi)*, and *Arabic (ar)* is also a source language, which is from different linguistic family. Following the setting in Cotterell & Duh (2017), we also simulate the low- and high-resource scenarios by creating 100 and 10,000 sentences split for training target language datasets, respectively. Then we create 1,000 sentences split for validation and test, respectively. For source languages, we create 10,000 sentence split for training only. For high-resource scenario, we only conduct experiments on *Galician (gl-high)* and *Ukrainian (uk-high)*. The list of selected datasets are described in Table 2.

Table 2: List of Named Entity Recognition Datasets in Pan et al. (2017).

| Language | Resource | Linguistic Family | Linguistic Branch | # Training Sentences | # Dev Sentences | # Test Sentences |
|---|---|---|---|---|---|---|
| Spanish (es) | Source | Indo-European | Romance | 10,000 | - | - |
| Galician (gl / gl-high) | Target | Indo-European | Romance | 100 / 10,000 | 1,000 | 1,000 |
| Dutch (nl) | Source | Indo-European | Germanic | 10,000 | - | - |
| West Frisian (fy) | Target | Indo-European | Germanic | 100 | 1,000 | 1,000 |
| Russian (ru) | Source | Indo-European | Slavic | 10,000 | - | - |
| Ukrainian (uk / uk-high) | Target | Indo-European | Slavic | 100 / 10,000 | 1,000 | 1,000 |
| Hindi (hi) | Source | Indo-European | Indo-Aryan | 10,000 | - | - |
| Marathi (mr) | Target | Indo-European | Indo-Aryan | 100 | 1,000 | 1,000 |
| Arabic (ar) | Source | Afro-Asiatic | Semitic | 10,000 | - | - |

## 4.2 Experimental Setup

We use 50-dimensional publicly available pre-trained word embeddings for English, Spanish and Dutch languages of CoNLL and WNUT datasets in our experiments, which are trained by word2vec package[2] on the corresponding Wikipedia articles (2017-12-20 dumps) (Lin et al., 2018). For the named entity datasets selected from Pan et al. (2017), we use 300-dimensional pre-trained word embeddings trained by fastText package[3] on Wikipedia (Bojanowski et al., 2017), and the 30-dimensional randomly initialized character embeddings are used for all the datasets. We set the filter number as 20 for char-level CNN and the dimension of hidden states of the word-level LSTM as 200 for both base model and DATNet-F. For DATNet-P, we set 100 for source, share, and target LSTMs dimension, respectively. Parameters optimization is performed by Adam optimizer (Kingma & Ba, 2014) with gradient clipping of 5.0 and learning rate decay strategy. We set the initial learning rate of $\beta_0 = 0.001$ for all experiments. At each epoch $t$, learning rate $\beta_t$ is updated using $\beta_t = \beta_0/(1 + \rho \times t)$, where $\rho$ is decay rate with 0.05. To reduce over-fitting, we also apply Dropout (Srivastava et al., 2014) to the embedding layer and the output of the LSTM layer, respectively.

## 4.3 Comparison with State-of-The-Art Results

In this section, we compare our approach with state-of-the-art (SOTA) methods on CoNLL and WNUT benchmark datasets. In the experiment, we exploit all the source data (i.e., CoNLL-2003 English NER) and target data to improve performance of target tasks. The averaged results with standard deviation over 10 repetitive runs are summarized in Table 3, and we also report the best results on each task for fair comparison with other SOTA methods. From results, we observe that incorporating the additional resource is helpful to improve performance. DATNet-P model achieves the highest F1 score on CoNLL-2002 Spanish and second F1 score on CoNLL-2002 Dutch dataset while DATNet-F model beats others on WNUT-2016 and WNUT-2017 English Twitter datasets. Different from other state-of-the-art models, DATNets do *not* use any addition features[4].

Table 3: Comparison with State-of-the-art Results in CoNLL and WNUT datasets (F1-score).

| Mode | Methods | | Additional Features | | | CoNLL Datasets | | WNUT Datasets | |
|---|---|---|---|---|---|---|---|---|---|
| | | | POS | Gazetteers | Orthographic | Spanish | Dutch | WNUT-2016 | WNUT-2017 |
| Mono-language /domain | Gillick et al. (2016) | | × | × | × | 82.59 | 82.84 | - | - |
| | Lample et al. (2016) | | × | √ | × | 85.75 | 81.74 | - | - |
| | Partalas et al. (2016) | | √ | √ | √ | - | - | 46.16 | - |
| | Limsopatham & Collier (2016) | | × | × | √ | - | - | 52.41 | - |
| | Lin et al. (2017a) | | √ | √ | × | - | - | - | 40.42 |
| | **Our Base Model** | Best | × | × | × | 85.53 | 85.55 | 44.96 | 35.20 |
| | | Mean & Std | | | | 85.35±0.15 | 85.24±0.21 | 44.37±0.31 | 34.67±0.34 |
| Cross-language /domain | Yang et al. (2017) | | × | √ | × | 85.77 | 85.19 | - | - |
| | Lin et al. (2018) | | × | √ | × | 85.88 | 86.55 | - | - |
| | Feng et al. (2018) | | √ | × | × | 86.42 | **88.39** | - | - |
| | von Däniken & Cieliebak (2017) | | × | √ | × | - | - | - | 40.78 |
| | Aguilar et al. (2017) | | √ | × | √ | - | - | - | 41.86 |
| | **DATNet-P** | Best | × | × | × | **88.16** | 88.32 | 50.85 | 41.12 |
| | | Mean & Std | | | | 87.89±0.18 | 88.09±0.13 | 50.41±0.32 | 40.52±0.38 |
| | **DATNet-F** | Best | × | × | × | 87.04 | 87.77 | **53.43** | **42.83** |
| | | Mean & Std | | | | 86.79±0.20 | 87.52±0.19 | 53.03±0.24 | 42.32±0.32 |

Table 4 summarizes the results of our methods under different cross-language transfer settings as well as the comparison with Cotterell & Duh (2017). In this experiment, we study the transferability between languages not only from same linguistic family and branch, but also from different linguistic families or branches. According to the results, DATNets outperform the transfer method of Cotterell & Duh (2017) for both low- and high-resource scenarios within the same linguistic family and branch (i.e., in-family in-branch) transfer case. We also observe that: 1) For the low-resource scenario, transfer learning is significantly helpful for improving the performance of target datasets

---

[2]https://github.com/tmikolov/word2vec

[3]https://github.com/facebookresearch/fastText

[4]Although our model performance on CONLL-2002 Dutch NER dataset is only comparable to the SOTA result, on the one hand, we do not use any addition features while the SOTA method did use; on the other, we are not sure if the SOTA method has incorporated the validation set into training. And if we merge training and validation sets, we can push the F1 score to 88.71, which outperforms the SOTA method.

within both same and different linguistic family or branch (i.e., in/cross-family in/cross-branch) transfer cases, while the improvements are more prominent under the in-family in-branch case. 2) For the high-resource scenario, say, when the target language data is sufficient, the improvements of transfer learning are not very distinct compared with that for low-resource scenario under in-family in-branch case. We also find that there is no effect by transferring knowledge from *Arabic* to *Galician* and *Ukrainian*. We suspect that it is caused by the great linguistic differences between source and target languages, since, for example, *Arabic* and *Galician* are from totally different linguistic families.

Table 4: Results of Varying Cross-language Transfer Settings in Pan et al. (2017) Datasets (F1-score).

| Language | | Transferring Strategy | Cotterell & Duh (2017) | | Our Methods | | |
| Source | Target | | Base Model | Transfer | Base Model | DATNet-P | DATNet-F |
|---|---|---|---|---|---|---|---|
| nl | fy | In-Family In-Branch | 58.43 | 72.12 | 57.47 | 75.08 | 76.05 |
| hi | fy | In-Family Cross-Branch | - | - | 57.47 | 69.25 | 68.44 |
| ar | fy | Cross-Family Cross-Branch | - | - | 57.47 | 67.89 | 66.05 |
| hi | mr | In-Family In-Branch | 39.02 | 60.92 | 43.55 | 68.55 | 64.87 |
| nl | mr | In-Family Cross-Branch | - | - | 43.55 | 63.83 | 60.50 |
| ar | mr | Cross-Family Cross-Branch | - | - | 43.55 | 63.28 | 59.76 |
| es | gl | In-Family In-Branch | 49.19 | 76.40 | 49.94 | 79.60 | 86.01 |
| hi | gl | In-Family Cross-Branch | - | - | 49.94 | 60.57 | 61.68 |
| ar | gl | Cross-Family Cross-Branch | - | - | 49.94 | 59.18 | 60.43 |
| es | gl-high | In-Family In-Branch | 89.42 | 89.46 | 92.78 | 93.14 | 93.02 |
| ar | gl-high | Cross-Family Cross-Branch | - | - | 92.78 | 92.63 | 92.21 |
| ru | uk | In-Family In-Branch | 60.65 | 76.74 | 61.48 | 79.02 | 80.76 |
| hi | uk | In-Family Cross-Branch | - | - | 61.48 | 72.73 | 73.84 |
| ar | uk | Cross-Family Cross-Branch | - | - | 61.48 | 71.55 | 72.24 |
| ru | uk-high | In-Family In-Branch | 87.39 | 87.42 | 93.29 | 93.62 | 93.51 |
| ar | uk-high | Cross-Family Cross-Branch | - | - | 93.29 | 92.83 | 92.42 |

[*] Base model means the model is trained by using target language dataset only.

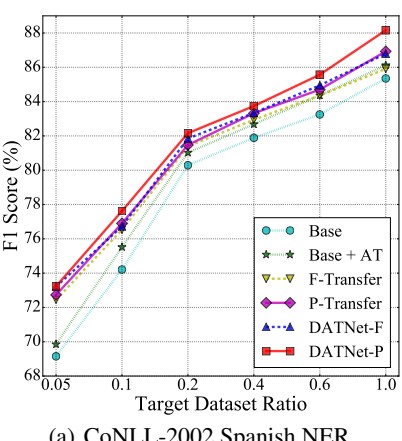

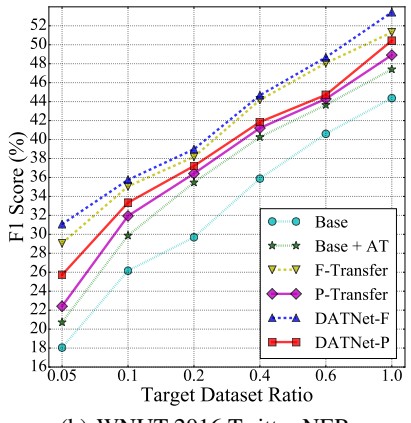

(a) CoNLL-2002 Spanish NER      (b) WNUT-2016 Twitter NER

Figure 2: Comparison with Different Target Data Ratio, where AT stands for adversarial training, F(P)-Transfer denotes the DATNet-F(P) without AT.

## 4.4 TRANSFER LEARNING PERFORMANCE

In this section, we investigate on improvements with transfer learning under multiple low-resource settings with partial target data. To simulate a low-resource setting, we randomly select subsets of target data with varying data ratio at 0.05, 0.1, 0.2, 0.4, 0.6, and 1.0. For example, 20,748 training tokens are sampled from the training set under a data ratio of $r = 0.1$ for the dataset CoNLL-2002 Spanish NER (Cf. Table 1). The results for cross-language and cross-domain transfer are shown in Fig. 2(a) and 2(b), respectively, where we compare the results with each part of DATNet under various data ratios. From those figures, we have the following observations: 1) both adversarial training and adversarial discriminator in DATNet consistently contribute to the performance improvement; 2) the transfer learning component in the DATNet consistently improve over the base model results and the improvement margin is more substantial when the target data

Table 5: Experiments on Extremely Low Resource (F1-score).

| Tasks | CoNLL-2002 Spanish NER | | | | | | WNUT-2016 Twitter NER | | | | | |
|---|---|---|---|---|---|---|---|---|---|---|---|---|
| # Target train sentences | 10 | 50 | 100 | 200 | 500 | 1000 | 10 | 50 | 100 | 200 | 500 | 1000 |
| Base | 21.53 | 42.18 | 48.35 | 63.66 | 68.83 | 76.69 | 3.80 | 14.07 | 17.99 | 26.20 | 31.78 | 36.99 |
| + AT | 19.23 | 41.01 | 50.46 | 64.83 | 70.85 | 77.91 | 4.34 | 16.87 | 18.43 | 26.32 | 35.68 | 41.69 |
| + P-Transfer | 29.78 | 61.09 | 64.78 | 66.54 | 72.94 | 78.49 | 7.71 | 16.17 | 20.43 | 29.20 | 34.90 | 41.20 |
| + F-Transfer | 39.72 | 63.00 | 63.36 | 66.39 | 72.88 | 78.04 | 15.26 | 20.04 | 26.60 | 32.22 | 38.35 | 44.81 |
| DATNet-P | 39.52 | 62.57 | 64.05 | **68.95** | **75.19** | **79.46** | 9.94 | 17.09 | 25.39 | 30.71 | 36.05 | 42.30 |
| DATNet-F | **44.52** | **63.89** | **66.67** | 68.35 | 74.24 | 78.56 | **17.14** | **22.59** | **28.41** | **32.48** | **39.20** | **45.25** |

ratio is lower. For example, when the data ratio is 0.05, DATNet-P model outperforms the base model by more than 4% absolutely in F1-score on Spanish NER and DATNet-F model improves around 13% absolutely in F1-score compared to base model on WNUT-2016 NER.

In the second experiment, we further investigate DATNet on the extremely low resource cases, e.g., the number of training target sentences is 10, 50, 100, 200, 500 and 1,000. The setting is quite challenging and fewer previous works have studied before. The results are summarized in Table 5. We have two interesting observations [5]: 1) DATNet-F outperforms DATNet-P on cross-language transfer when the target resource is extremely low, however, this situation is reversed when the target dataset size is large enough (here for this specific dataset, the threshold is 100 sentences); 2) DATNet-F is always superior to DATNet-P on cross-domain transfer. For the first observation, it is because DATNet-F with more shared hidden units is more efficient to transfer knowledge than DATNet-P when data size is extremely small. For the second observation, because cross-domain transfer are in the same language, more knowledge is common between the source and target domains, requiring more shared hidden features to carry with these knowledge compared to cross-language transfer. Therefore, for cross-language transfer with an extremely low resource and cross-domain transfer, we suggest using DATNet-F model to achieve better performance. As for cross-language transfer with relatively more training data, DATNet-P model is preferred.

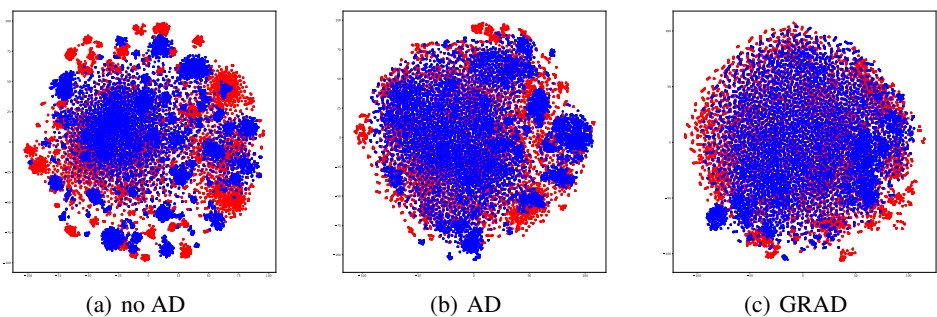

|  (a) no AD  |  (b) AD  |  (c) GRAD  |

Figure 3: The visualization of extracted features from shared bidirectional-LSTM layer. The left, middle, and right figures show the results when no Adversarial Discriminator (AD), AD, and GRAD is performed, respectively. Red points correspond to the source CoNLL-2003 English examples, and blue points correspond to the target CoNLL-2002 Spanish examples.

## 4.5 ABLATION STUDY OF DATNET

In the proposed DATNet, both GRAD and AT play important roles in low resource NER. In this experiment, we further investigate how GRAD and AT help transfer knowledge across language/domain. In the first experiment[6], we used t-SNE (Maaten & Hinton, 2008) to visualize the feature distribution of BiLSTM outputs without AD, with normal AD (GRAD without considering data imbalance), and with the proposed GRAD in Figure 3. From this figure, we can see that the GRAD in DATNet makes the distribution of extracted features from the source and target datasets

---

[5] For other tasks/languages we have the similar observation, we only report CoNLL-2002 Spanish and WNUT-2016 Twitter results due to the page limit.

[6] We used data ratio $\rho = 0.5$ for training model and randomly selected 10k testing data for visualization.

much more similar by considering the data imbalance, which indicates that the outputs of BiLSTM are resource-invariant.

Table 6: Quantitative Performance Comparison between Models with Different Components.

| CoNLL-2002 Spanish NER | | | | WNUT-2016 Twitter NER | | | |
|---|---|---|---|---|---|---|---|
| Model | F1-score | Model | F1-score | Model | F1-score | Model | F1-score |
| Base | 85.35 | +AT | 86.12 | Base | 44.37 | +AT | 47.41 |
| +P-T (no AD) | 86.15 | +AT +P-T (no AD) | 86.90 | +P-T (no AD) | 47.66 | +AT +P-T (no AD) | 48.44 |
| +F-T (no AD) | 85.46 | +AT +F-T (no AD) | 86.17 | +F-T (no AD) | 49.79 | +AT +F-T (no AD) | 50.93 |
| +P-T (AD) | 86.32 | +AT +P-T (AD) | 87.19 | +P-T (AD) | 48.14 | +AT +P-T (AD) | 49.41 |
| +F-T (AD) | 85.58 | +AT +F-T (AD) | 86.38 | +F-T (AD) | 50.48 | +AT +F-T (AD) | 51.84 |
| +P-T (GRAD) | 86.93 | +AT +P-T (GRAD) (*DATNet-P*) | **88.16** | +P-T (GRAD) | 48.91 | +AT +P-T (GRAD) (*DATNet-P*) | 50.85 |
| +F-T (GRAD) | 85.91 | +AT +F-T (GRAD) (*DATNet-F*) | 87.04 | +F-T (GRAD) | 51.31 | +AT +F-T (GRAD) (*DATNet-F*) | **53.43** |

\* AT: Adversarial Training;  P-T: P-Transfer;  F-T: F-Transfer;  AD: Adversarial Discriminator; GRAD: Generalized Resource-Adversarial Discriminator.

To better understand the working mechanism, Table 6 further reports the quantitative performance comparison between models with different components. We observe that GRAD shows the stable superiority over the normal AD regardless of other components. There are no always winner between DATNet-P and DATNet-F on different settings. DATNet-P architecture is more suitable to cross-language transfer while DATNet-F is more suitable to cross-domain transfer.

Table 7: Analysis of Maximum Perturbation $\epsilon_{\mathbf{w}_T}$ in AT with Varying Data Ratio $\rho$ (F1-score).

| $\epsilon_{\mathbf{w}_T}$ | 1.0 | 3.0 | 5.0 | 7.0 | 9.0 |
|---|---|---|---|---|---|
| Ratio | | CoNLL-2002 Spanish NER | | | |
| $\rho = 0.1$ | 75.90 | 76.23 | 77.38 | 77.77 | **78.13** |
| $\rho = 0.2$ | 81.54 | 81.65 | 81.32 | **81.81** | 81.68 |
| $\rho = 0.4$ | 83.62 | 83.83 | 83.43 | **83.99** | 83.40 |
| $\rho = 0.6$ | 84.44 | 84.47 | **84.72** | 84.04 | 84.05 |

From the previous results, we know that AT helps enhance the overall performance by adding perturbations to inputs with the limit of $\epsilon = 5$, i.e., $\|\eta\|_2 \leq 5$. In this experiment, we further investigate how target perturbation $\epsilon_{\mathbf{w}_T}$ with fixed source perturbation $\epsilon_{\mathbf{w}_S} = 5$ in AT affects knowledge transfer and the results on Spanish NER are summarized in Table 7. The results generally indicate that less training data require a larger $\epsilon$ to prevent over-fitting, which further validates the necessity of AT in the case of low resource data.

Table 8: Analysis of Discriminator Weight $\alpha$ in GRAD with Varying Data Ratio $\rho$ (F1-score).

| $\alpha$ | 0.1 | 0.15 | 0.2 | 0.25 | 0.3 | 0.35 | 0.4 | 0.45 | 0.5 | 0.55 | 0.6 | 0.65 | 0.7 | 0.75 | 0.8 |
|---|---|---|---|---|---|---|---|---|---|---|---|---|---|---|---|
| Ratio | | | | | | | CoNLL-2002 Spanish NER | | | | | | | | |
| $\rho = 0.1$ | 78.37 | 78.63 | **78.70** | 78.32 | 77.96 | 77.92 | 77.88 | 77.78 | 77.85 | 77.90 | 77.65 | 77.57 | 77.38 | 77.49 | 77.29 |
| $\rho = 0.2$ | 80.99 | 81.71 | **82.18** | 81.57 | 81.53 | 81.55 | 81.44 | 81.25 | 81.32 | 81.16 | 81.02 | 81.16 | 80.63 | 80.79 | 80.54 |
| $\rho = 0.4$ | 83.76 | 83.73 | 84.18 | **84.48** | 84.26 | 84.12 | 83.54 | 83.40 | 83.52 | 84.18 | 83.42 | 83.47 | 83.28 | 83.33 | 83.19 |
| $\rho = 0.6$ | 85.18 | 85.24 | 85.85 | 85.68 | 85.84 | **86.10** | 85.71 | 85.74 | 85.42 | 85.60 | 85.20 | 85.40 | 85.26 | 85.24 | 84.98 |

Finally, we analyze the discriminator weight $\alpha$ in GRAD and results are summarized in Table 8. From the results, it is interesting to find that $\alpha$ is directly proportional to the data ratio $\rho$, basically, which means that more target training data requires larger $\alpha$ (i.e., smaller $1 - \alpha$ to reduce training emphasis on the target domain) to achieve better performance.

## 5 CONCLUSION

In this paper we develop a transfer learning model DATNet for low-resource NER, which aims at addressing two problems remained in existing work, namely representation difference and resource data imbalance. We introduce two variants of DATNet, DATNet-F and DATNet-P, which can be chosen for use according to the cross-language/domain user case and the target dataset size. To improve model generalization, we propose dual adversarial learning strategies, i.e., AT and GRAD. Extensive experiments show the superiority of DATNet over existing models and it achieves new state-of-the-art performance on CoNLL NER and WNUT NER benchmark datasets.

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
