# OpenReview forum: "DATNet: Dual Adversarial Transfer for Low-resource Named Entity Recognition"
_ICLR.cc/2019/Conference_

### Official Review · AnonReviewer3 · 2018-11-02
**Interesting model, thorough exposition and analysis, but partly flawed experiment setup**

**Rating:** 6
**Confidence:** 4

**Review:**

The paper introduces a novel architecture for low-resource named entity tagging: a dual adversarial transfer network, in which fusion between high- and low-resource, or high- and low-noise data is achieved via also novel resource-adversarial discriminator.

The model is interesting, novel, clearly exposed in sufficient detail, and warrants publication as such. The idea to unify representation differences and data imbalance under one model is noteworthy.

I find that the description of related work, especially in the introduction, does not credit past contributions sufficiently. For one, large parallel corpora do exist for many languages, albeit some of them may not be sufficiently ample in named entities to facilitate cross-lingual NER. Yet, for the fortunate ones, such corpora do make for rather reasonable NER taggers via multi-source projection (cf. Enghoff et al., W-NUT 2018). Absent is the prominent work by Mayhew et al. (2017) in cross-lingual NER, as well as Pan et al. (2017) who engage with evaluation in 282 languages.

This unfair account of related work would not trouble me as much if it weren't coupled with an experiment in "low-resource" NER that features---Spanish and Dutch as target languages. Firstly, these languages are rich in resources, after all, they featured in CoNLL 2003, for one. Secondly, they are closely related to English as the source language, and any simulated low-resource scenario that features both the injection of target-language data *and* a very closely related source language is simply *not* representative of any true low-resource scenario.

This experiment setup troubles me, especially in light of real and synthetic NER data available to test the setup for true low-resource languages: from silver data by Al-Rfou et al. (2015) or Pan et al. (2017), via Mayhew et al. (2017) or Cotterrell and Duh (2017) who test on 10-15 gold datasets, etc., real low-resource NER data that is multilingual can be found. Any paper that in 2018 claims to do low-resource NER and then simulates a setup with Dutch and Spanish is poor scholarship in my submission, regardless of the clever model.

I do let the clever model upvote my review, but not beyond borderline.

Minor:
- the use of "lingual" as noun is rather off-putting, at least to me

---

> ### Author Response · Authors · 2018-11-20
> **Response to Reviewer 3**
>
>
> Comment 1:  "I find that the description of related work, especially in the introduction, does not credit past contributions sufficiently. For one, large parallel corpora do exist for many languages, albeit some of them may not be sufficiently ample in named entities to facilitate cross-lingual NER. Yet, for the fortunate ones, such corpora do make for rather reasonable NER taggers via multi-source projection (cf. Enghoff et al., W-NUT 2018). Absent is the prominent work by Mayhew et al. (2017) in cross-lingual NER, as well as Pan et al. (2017) who engage with evaluation in 282 languages."
>
> Response 1: Thanks for pointing out the missing references. We have updated the related work section as suggested.
>
> ---------------------------------------------------------------------------------------------------------------------------------------------------------------------
> Comment 2:  "This unfair account of related work would not trouble me as much if it weren't coupled with an experiment in "low-resource" NER that features---Spanish and Dutch as target languages. Firstly, these languages are rich in resources, after all, they featured in CoNLL 2003, for one. Secondly, they are closely related to English as the source language, and any simulated low-resource scenario that features both the injection of target-language data *and* a very closely related source language is simply *not* representative of any true low-resource scenario.
> This experiment setup troubles me, especially in light of real and synthetic NER data available to test the setup for true low-resource languages: from silver data by Al-Rfou et al. (2015) or Pan et al. (2017), via Mayhew et al. (2017) or Cotterrell and Duh (2017) who test on 10-15 gold datasets, etc., real low-resource NER data that is multilingual can be found. Any paper that in 2018 claims to do low-resource NER and then simulates a setup with Dutch and Spanish is a poor scholarship in my submission, regardless of the clever model."
>
> Response 2: Thanks for the suggestion. In the revision, we further conducted the experiment on a  large-scale cross-lingual named entity dataset in [Pan  et al.,2017] which contains 282 languages for evaluation.  We chose 9 languages (Galician, West Frisian,  Ukrainian,  Marathi,  Spanish,  Dutch,  Russian,  Hindi,  Arabic) in our experiments given their low resource-nature and different linguistic nature. We follow the same setting as [Cotterrell and Duh (2017)] to simulate high- and low- resource scenarios. Table 4 summarizes the results of our methods under different cross-lingual transfer settings as well as the comparison with Cotterrell and Duh (2017). From the results,  we can clearly observe the superiority of our proposed model over the state-of-the-art, which can demonstrate that our model can really transfer from high-resource NER data to low-resource.
>
> ---------------------------------------------------------------------------------------------------------------------------------------------------------------------
> Comment 3:  “the use of "lingual" as noun is rather off-putting, at least to me”
>
> Response 3: Thank you for this comment. In the revision, we replaced the term "lingual" with "language".

---

### Official Review · AnonReviewer2 · 2018-11-03
**Interesting approach for cross-domain / cross-lingual NER with solid empirical results; limited technical novelty**

**Rating:** 6
**Confidence:** 5

**Review:**

<Summary>
Authors propose the new “DATNet” for the NER task, which extends the base neural model for NER (Bi-LSTM+CRF sequence model with input represented with CharCNN-word embeddings) with the following two main components: (1) GRAD: a language (or resource) adversarial discriminator with adaptive weights that regularize source-target data imbalance, and (2) additional adversarial training approaches that perturb input samples in the embeddings space.

The paper reports big improvement over their baseline approaches without having to rely on other auxiliary or hand-crafted features. The experiment is performed for various low-resource scenarios (varying training data size).

<Comments>
- While the idea of applying “dual adversarial” approaches is new in the context of NER, the technical novelty of each component is limited. GRAD, for example, is rather a minor modification of Language Adversarial Discriminator (Kim et al., 2017) with a scalar weight parameter on loss. The empirical superiority of the proposed method (GRAD) over normal AD approaches cannot be claimed either -- for this authors need to report quantitative results of the (Base + AT + F/P-transfer with AD -- or e.g. \alpha fixed at 0.5 or at some other rate), which I believe is missing in all figures and tables. Visualization of resulting feature distribution (Figure 3) is interesting to look at, but that alone does not suffice. There is no technical novelty in applying adversarial training either, except that it was used in the context of NER.


- Authors use both source and target data to train their “base model” as well (“... we exploit all the source data and target data ...“), e.g. presumably by merging the source and target dataset as well as their embedding matrices, etc. It is important that authors report if there ever is a negative transfer case (e.g. a base model trained with just target data may outperform models trained with source+target data) at varying resource scenarios -- especially at sufficiently-resourced cases.
If by any chance their “base model” refers to in-domain training / in-domain testing results on target, the aforementioned baseline (naive merge of source and target data) is obviously an important baseline to report. I suggest that authors provide these details or clarify. (The confusion comes mainly because some of the SOTA results the authors quote are in-domain training / in-domain evaluation results on respective languages, and some are cross-domain results -- yet they are all under “cross-lingual/domain” columns in Table 2).

- It would be interesting to report the learned optimal \alpha value for each different setting (at varying r or training size) to see if the authors’ intuition is met. On a related note, from the manuscript alone, it is not entirely clear if \alpha is a learnable parameter or a tunable hyper parameter -- by context I believe it is a model parameter. If they are tuned, authors need to report these values.

<Nit>
- Section 3.2.3, “... (GRAD) to enable adaptive weights for [each sample]” → I think it reads better with [each resource] or [each source and target], unless you meant \alpha_i for each sentence.
- Section 3.2.5, “... recently, adversarial samples are [wisely] incorporated” → [widely]
- Fonts for figures could be bigger.

---

> ### Author Response · Authors · 2018-11-20
> **Response to Reviewer 2**
>
>
> Comment 1: “...the technical novelty of each component is limited. GRAD, for example, is rather a minor modification of Language Adversarial Discriminator (Kim et al., 2017) with a scalar weight parameter on the loss. … for this authors need to report quantitative results of the (Base + AT + F/P-transfer with AD...”
>
> Response 1：Thanks for your comments. The major difference between our GRAD and that proposed by (Kim et al, 2017) is that we introduced the source weight \alpha and instance weight \gamma to address the influence of imbalanced training data and dominant easy training samples. The instance weight is adaptively learned automatically from data, which has not been explored in related literature to the best of our knowledge. The effectiveness of these components with elegant and reasonable modeling is shown in Table 3 by comparing our performance with other state-of-the-art.  We also added a more quantitative comparison of model F/P-transfer without AD,  with AD,  and with GRAD  in Table 6.  From the results, we found that GRAD consistently outperforms AD across different settings.
>
> -------------------------------------
> Comment 2:  "Authors use both source and target data to train their “base model” as well …e.g. presumably by merging the source and target dataset as well as their embedding matrices, etc."
>
> Response 2:  Thank you for pointing out this misunderstanding. Our base model only uses the target data.  We have clarified this in the footnote of Table  4.
>
> ------------------------------------
> Comment 3:  "It is important that authors report if there ever is a negative transfer case … at varying resource scenarios -- especially at sufficiently-resourced cases."
>
> Response 3: Thanks for the suggestion. In the revision,  we also experiment on a large-scale cross-lingual named entity dataset which was brought out by (Pan  et al.,2017)  and contains 282 languages for evaluation. We choose 9 languages in our experiments and the results are summarized in Table 4.  We observe that for the high-resource scenario, say, when the target language data is sufficient, the improvements of transfer learning are not very distinct compared with that for low-resource scenario under in-family in-branch case,  and we also find that there is no effect by transferring knowledge from Arabic to Galician and Ukrainian, which we suspect is caused by the great linguistic differences between source and target languages, since, for example, Arabic and Galician are from totally different linguistic families.
>
> --------------------------------
> Comment 4: “the aforementioned baseline (naive merge of the source and target data) is obviously an important baseline to report.”
>
> Response 4: Training with the naive merge of the source and target data is equivalent to our baseline (Base model +  F-T  without AD) shown in Table 6.
>
> --------------------------------
> Comment 5:  "The confusion comes mainly because some of the SOTA results the authors quote are in-domain training / in-domain evaluation results on respective languages, and some are cross-domain results -- yet they are all under “cross-lingual/domain” columns in Table 2".
>
> Response 5:  Thanks for your suggestion.  We have updated the Table  3 in the revision with the separation for the mono-language/domain and cross-language/domain.
>
> ----------------------------
> Comment 6: " It would be interesting to report the learned optimal \alpha value for each different setting (at varying r or training size) to see if the authors’ intuition is met."
>
> Response 6:  Yes,  \alpha is tunable.  We analyze the discriminator weight \alpha in GRAD and results are summarized in Table  8. From the results, it is interesting to find that \alpha is directly proportional to the data ratio \rho, basically, which means that more target training data requires larger \alpha (i.e., smaller 1-\alpha to reduce training emphasis on the target domain) to achieve better performance.
>
> -----------------------------
> Comment 7:  "- Section 3.2.3, “... (GRAD) to enable adaptive weights for [each sample]” → I think it reads better with [each resource] or [each source and target] unless you meant \alpha_i for each sentence."
>
> Response 7: Here \alpha is indeed adapted for each sentence. We have clarified it in the corresponding place in the revision like this: "To overcome this issue, we further propose Generalized Resource-Adversarial Discriminator (GRAD) to enable adaptive weights for each sample (note that the sample here means each sentence of resource), which focuses the model training on hard samples."
>
> ---------------------------
> Comment 8:  "Section 3.2.5, “... recently, adversarial samples are [wisely] incorporated” → [widely]"
>
> Response 8: Thanks for this comment. We have replaced "wisely" with "widely".
>
> ---------------------------
> Comment 9: "Fonts for figures could be bigger."
>
> Response 9: Thank you for this suggestion. We have enlarged the font of figures for clearness in the revision.

---

### Official Review · AnonReviewer1 · 2018-11-11
**Nicely and clear written paper with innovative aspects. The results look strong but could increase confidence in results via reporting of variance / stat sig; Paper is a bit narrow as applied new method to one task only.**

**Rating:** 6
**Confidence:** 4

**Review:**

The authors propose a new architecture Dual Adversarial Transfer Network for addressing low-resource NER. They achieve a new SOTA on low resource language. The authors compare a base-line with two alternatives based on variants of GANs.

The results go beyond SOTA for low resource NER which seems a solid contribution. The paper is well and clearly written and I would be able to replicate the experiments. I wonder if this is a new architecture that works well for one task or if it could be applied to other tasks too. This would strengthen the approach and paper quite a bit. Could the method for instance be applied to other labeling tasks: POS tagging, morphological features. This would increase the potential impact substantially.

In low resource scenarios often methods work that do stop working at a some point with more resources. For this methods the boundary when this effect occurs would be interesting to explore; Figure 2 goes into this direction which is a quite nice study but the boundary is not  explored further; by using for instance the English NER data. Additionally, the performance on the English data set would indicate what the method could perform in comparison to current SOTA for normal resource setting - you could use some of the low resources in addition. The English data set was used but only to exploit it for the transfer learning.
Table 2 is a good overview on SOTA. I really wonder about the variance of the results of the system which can be depending on the network quite large. Why not running repetition test, this  would enable the authors to report variance and statistically significance between the baseline and their other systems.
I wonder also how more standard exploitation of additional data would do such as Bert, ELMO or older methods such as up-training - this would help to get a more complete picture and strength the paper further.

The paper could be stronger by applying the method to other task too as stated the authors - this is a ‘new architectures’ (for NER) which triggers the question and does it generalize to other tasks? In the conclusion there is even the claim as a statement! that this can be generalized to other NLP task without actually trying. I think, this can not be claimed in the conclusions without pursuing this in some other task and I suggest to tune this down.

Overall:
Nicely and clear written paper containing innovative elements. The results look strong too me but due to the lack of variance and stat sig., I am not sure if they are really super strong. The paper could be stronger by applying the method to other task too as stated ‘new architectures’ which triggers the question if the method generalizes to other tasks?

---

> ### Author Response · Authors · 2018-11-20
> **Response to Reviewer 1**
>
> Comment 1:  "...Could the method, for instance, be applied to other labeling tasks: POS tagging, morphological features... "
> Response  1:  Yes, our proposed architecture is easy to be applied to any other sequence labeling task.  However, the scope of this paper is focused on the NER task, which is the most challenging and representative task among all the sequence labeling tasks. And we appreciate this suggestion and plan to extend this method to more related tasks in the future.
> -------------------------------------------------------------------------------------------------------------------------------------------------------------
> Comment 2: "In low resource scenarios often methods work that do stop working at some point with more resources. For this methods the boundary when this effect occurs would be interesting to explore; Figure 2 goes into this direction which is a quite nice study but the boundary is not explored further; by using, for instance, the English NER data. Additionally, the performance on the English dataset would indicate what the method could perform in comparison to current SOTA for normal resource setting - you could use some of the low resources in addition. The English dataset was used but only to exploit it for the transfer learning. "
>
> Response  2:  The target of this paper is to transfer the knowledge obtained from high-resource language to the low-resource language so that we can boost the performance of low-resource NER, which is exactly what transfer learning aims at. Trying to beat the state-of-the-art high-resource NER such as the English NER (even with the help of other low-resources) is not within the scope of this work. Indeed, in order to further validate the effectiveness of our model on transfer learning of low-resources NER, we experimented on 9 more low-resources languages and proved that our model outperforms the state-of-the-art across the in/cross-family and in/cross-branch scenarios.
> -------------------------------------------------------------------------------------------------------------------------------------------------------------
> Comment 3: "Table 2 is a good overview of SOTA. I really wonder about the variance of the results of the system which can be depending on the network quite large. Why not running repetition test, this would enable the authors to report variance and statistically significance between the baseline and their other systems."
>
> Response  3:  Yes,  in the revision version,  we updated the Table 3 with the averaged F1-scores and the standard deviation over 10 repetitive runs on each task.
> -------------------------------------------------------------------------------------------------------------------------------------------------------------
> Comment 4:  “I wonder also how more standard exploitation of additional data would do such as Bert, ELMO or older methods such as up-training - this would help to get a more complete picture and strength the paper further.”
>
> Response 4:  In this paper, we aim at exploring the effectiveness of our proposed model, which is orthogonal to those methods that utilize the additional external data. That is to say, it is easy to apply the methods exploiting the external unlabeled data such as ELMO to our system, which for sure can strength the performance. But the scope of this work is still to demonstrate that our proposed transfer learning architecture can better and more efficient transfer knowledge from high-resources to low-resources. Overall, we thank you for this good suggestion, and we leave the exploitation of additional data for future work.
> -------------------------------------------------------------------------------------------------------------------------------------------------------------
> Comment 5: “The paper could be stronger by applying the method to other tasks too as stated the authors - this is a ‘new architectures’ (for NER) which triggers the question and does it generalize to other tasks? In conclusion, there is even the claim as a statement! that this can be generalized to other NLP task without actually trying. I think this can not be claimed in the conclusions without pursuing this in some other task and I suggest to tune this down.”
>
> Response 5: Thank you so much for mentioning this over-statement out. We have now removed the corresponding statement in the conclusion section. As aforementioned, we plan to generalize the proposed system to other NLP tasks to further validate its effectiveness on a more broad range of NLP problems in the future work.

---

### Meta-Review · Area_Chair1 · 2018-12-14
**Reject**

**Confidence:** 4
**Recommendation:** Reject

**Metareview:**

A focused contribution that is clearly presented. That being said, the task of low-resource named entity recognition is fairly narrow and it is hard to tell how significant the empirical results are. The paper could be much stronger if it evaluated on a second task (and third task). Right now it is unclear whether the technique would generalize to other tasks.